# INDIVIDUAL FAIRNESS AS AN EXTENSION OF GROUP FAIRNESS

## ABSTRACT

Since its formal definition in 2011, *individual fairness* has received relatively little attention from the machine learning community in comparison to *group fairness*. The reasons for this are several-fold. In order to implement it, one must define a similarity metric; obtaining this is a non-trivial task and an active research area. According to individual fairness, discontinuity in modelling, and thus deterministic classification, is inherently unfair. To achieve individual fairness we must turn to probabilistic models in which predictions are randomised. For many, this flies in the face of logic. Perhaps most importantly, researchers have conflicting views on its compatibility with group fairness.

In this work we attempt to address conflicting research on the nature of individual fairness. We clarify important defining features of individual fairness, framing it as an extension of group fairness, rather than acting in opposition to it. We review empirical evidence of the trade-off between group and individual fairness and analyse the associated individual fairness metric (which we term *individual cost*). Taking an analytical approach, we derive a new representation for individual cost in terms of model accuracy and expected error. With this representation we are able to apply finite difference method to identify the deviation region, that is, the cases in which individual cost conflicts with model accuracy. We conclude that empirical evidence does not support the existence of a trade-off between group and individual fairness but rather, likely demonstrates the well known trade-off between fairness and utility.

## 1 INTRODUCTION

The proliferation of data driven algorithmic solutions in social domains has been a catalyst for research in fair machine learning in recent years. Applications in high stakes decisions for criminal justice, predictive policing Ensign et al. (2018), healthcare, finance and beyond, have fueled the need for formal definitions of fairness notions, metrics, and mitigation techniques. In a recent survey, on fairness in machine learning, authors highlight five major dilemmas regarding progress in the space. The first two of these concern trade-offs. The first is that between fairness and model performance Hajian & Domingo-Ferrer (2012); Corbett-Davies et al. (2017); Calmon et al. (2017); Haas (2019). The second is that between different notions of fairness Darlington (1971); Chouldechova (2016); Kleinberg et al. (2016); Hardt et al. (2016); Barocas et al. (2019); Murgai (2023). The latter of these is credited with stifling progress in earlier research in the space Cole & Zieky (2001); Hutchinson & Mitchell (2019). Thus clarity around compatibility of different fairness measures of performance and fairness are important in moving the field forward.

A common claim is that there is a trade-off/tension between the notions of *individual fairness* and *group fairness* Narayanan (2018); Speicher et al. (2018); Pessach & Shmueli (2022); Caton & Haas (2023). This stems from the concern that, in cases where the target trait is indeed distributed differently between sensitive subgroups, the only way to satisfy statistical parity, is through preferential treatment of one group over another; but this in turn would violate the requirement to treat similar individuals similarly. For this case, Dwork et al. themselves suggest a modification that relaxes the Lipschitz condition Dwork et al. (2011). More recently Binns argued the conflict lies in the implementation rather than the concepts Binns (2019).

Apart from the above scenario, the claim of a trade-off between group and individual fairness has empirical support in Speicher et al. (2018) and it is this which serves as motivation for studying the corresponding fairness metric (referred to as *individual cost* hereafter). In this paper, we highlight important differences in the definition of individual fairness as originally defined by Dwork et al. (2011) and in the empirical analysis by Speicher et al. (2018). In particular, we show that while individual fairness defined by Dwork et al. can be viewed as an extension of group fairness, that decouples the tasks of ensuring fairness and maximising utility, individual cost (as the choice of name

implies) can be viewed as an extension of utility (or rather expected risk) and so is in some cases is a monotonic function of accuracy.

The definition of individual cost in the original paper is extremely flexible, and authors make several seemingly arbitrary choices for parameters. Thus, while the construction of the metric seems principled, its behaviour is opaque. Nevertheless, individual cost has been implemented in open source libraries Bellamy et al. (2018) and has traction among researchers. Recent surveys include it Pessach & Shmueli (2022); Caton & Haas (2023) and one recent paper optimises for it while developing a fair algorithm Jin et al. (2023).

Here we examine individual cost in detail, shedding light on its behaviour and meaning. We reason about its generality and describe constraints on parameters that ensure it is not a measure of expected risk, but rather *expected luck*. In addition we discuss important properties of the metric for different choices of the generalisation parameter $\alpha$. In the last part of the paper, we derive a new representation of individual cost under the assumption that accurate predictions are equally lucky for the individual. This allows us to visualise the behaviour of individual cost, covering all but one remaining degree of freedom. Finally we employ finite difference methods in this novel context in order to analytically describe the *deviation region*. That is, the part of the model solution space where incrementally increasing accuracy, increases unfairness.

The rest of this paper is organised as follows. In Section 2 we briefly review notions of fairness relevant to the discussion. In Section 3 we move on to individual fairness Dwork et al. (2011), clarifying its definition and position as an extension of group fairness, rather than being in contention with it. We note important conceptual features, and pay special attention to the case of a binary classifier, as this is the case for which empirical evidence of a trade-off is claimed Speicher et al. (2018). In Section 4, we describe individual cost Speicher et al. (2018). In section 5, we derive a new representation of individual cost, showing it can be written as a function of two model performance related parameters; namely, prediction accuracy and mean prediction error. With this new representation of the metric, we are able to answer the following question. For which set of model error distributions does incrementally increasing the accuracy, reduce fairness according to this metric? Our analysis shows that this only happens when the ratio of false positives to correct predictions is sufficiently high (greater than half the number of accurate predictions for the specific parameter choices in Speicher et al. (2018)). Furthermore, such a ratio can only be achieved if the prediction accuracy is sufficiently low (66.7% for the specific parameter choices in Speicher et al. (2018)). We conclude that the empirical evidence does not support the existence of a trade-off between group and individual fairness but rather demonstrates the well known trade-off between fairness and utility.

Our main contibution is to elucidate the behaviour of individual fairness and individual cost.

- We clarify important defining features of individual fairness, framing it as an extension of group fairness rather than acting in opposition to it.
- We conduct a detailed analysis of *individual cost*, showing how it can be viewed as an extension of expected risk. In addition we discuss some much needed guidance around parameter selection in calculating individual cost highlighting index properties for different choices.
- We derive a new representation of individual cost in terms of model performance metrics and reconcile conflicting empirical evidence demonstrating a trade-off between group and individual fairness.
- Using finite difference method, we derive an analytical expression for the deviation region - where decreasing accuracy increases fairness.

## 2 Background

Early methods for quantifying fairness, motivated by the introduction of anti-discrimination laws in the US in the 1960s, were developed with the goal of measuring bias in assessment tests for education and employment opportunities Cleary (1968); Einhorn & Bass (1971); Cole (1973); Novick & Petersen (1976). The metrics presented in these works fall under what has since become known as *group fairness* Barocas et al. (2019); Hutchinson & Mitchell (2019); Pessach & Shmueli (2022); Caton & Haas (2023) which considers differences in treatment (outcomes or errors) across subgroups of a population. Informally, *individual fairness* is the notion that similar individuals should be treated similarly Aristotle (350 B.C.); Guion (1966). A formal definition as equivalence to continuity with capped gradient, was first proposed by Dwork et al. (2011). Shortly after Zemel et al. Zemel et al. (2013) introduced their individual fairness metric which quantifies the *consistency* of a mapping, by comparing each individual outcome with the mean of the outcomes for its $k$ nearest neighbours and averaging over the population. More recently Speicher et al. Speicher et al.

(2018) proposed a measure of individual fairness, which we term individual cost, and is the subject of much of this paper. Finally, related to both group and individual fairness is the concept of counterfactual fairness Kilbertus et al. (2017); Kusner et al. (2018), which argues that the prediction distribution should be independent of both the sensitive features, and their causally dependent attributes.

In 2011, Dwork et. al. moved the formal definition of fairness beyond discrimination based on protected characteristics, towards a more generalised definition known as *individual fairness* Dwork et al. (2011). Individual fairness is based on the notion that similar individuals (with respect to the task) should be treated similarly and does not restrict its attention to protected characteristics, but allows all features to be incorporated in the notion of fairness. Its origin can be traced back to Aristotle, who wrote in the context of legal theory that "like cases should be treated alike" Schauer (2018). Formal definition of the concept came in the form of a continuity constraint on the model mapping. This new definition resolved the above issues with group fairness by construction. Importantly, by requiring the definition of a similarity metric, it decoupled the tasks of maximising utility and ensuring fairness so that they can be performed independently.

Implementation of individual fairness presents several challenges, the main one being that its definition is incomplete. One must define a similarity metric in feature space. Researchers and practitioners alike argue that rather than solving the problem of finding a fair model that maps individuals to predictions, individual fairness has added to the challenge by requiring the definition of a task specific similarity metric. Indeed, obtaining the similarity metric is a non-trivial task and an active area of research Zemel et al. (2013); Lahoti et al. (2019). We argue that this new paradigm makes explicit the importance of consistency in fair decision making. According to individual fairness, which requires continuity Dwork et al. (2011), deterministic classification is inherently unfair Narayanan (2018) and to achieve it, we must turn to probabilistic models in which predictions are randomised Dwork et al. (2011). For many, this flies in the face of logic.

## 3   INDIVIDUAL FAIRNESS

Informally, given a model that maps individuals to predictions, individual fairness can be interpreted as the requirement that, two individuals that are close in input (feature) space $\mathcal{X}$ are also close in output (target/prediction) space $\mathcal{Y}$. Dwork et al. note that in order to satisfy this constraint, the model mapping must be continuous, as defined next.

**Lipschitz Continuity**   Consider a model to be a function $\hat{y} = f(\boldsymbol{x})$, which maps individuals $\boldsymbol{x} \in \mathcal{X}$ to probability distributions over $y \in \mathcal{Y}$, that is, $f : \mathcal{X} \mapsto \mathcal{P}(\mathcal{Y})$. Then the mapping $f$ is *Lipschitz continuous* if there exists a real valued, non-negative constant $K \in \mathbb{R}_{\geq 0}$ such that

$$d_{\mathcal{P}(\mathcal{Y})}(f(\boldsymbol{x}_i), f(\boldsymbol{x}_j)) \leq K d_{\mathcal{X}}(\boldsymbol{x}_i, \boldsymbol{x}_j) \quad \forall \, \boldsymbol{x}_i, \boldsymbol{x}_j \in \mathcal{X}$$

where $d_{\mathcal{X}} : \mathcal{X} \times \mathcal{X} \mapsto \mathbb{R}$ and $d_{\mathcal{P}(\mathcal{Y})} : \mathcal{P}(\mathcal{Y}) \times \mathcal{P}(\mathcal{Y}) \mapsto \mathbb{R}$ denote distance metrics. $d_{\mathcal{X}}$ determines how far apart two individuals are in feature space and $d_{\mathcal{P}(\mathcal{Y})}$ measures the distance between two probability distributions over $\mathcal{Y}$.

For individual fairness we can omit the Lipschitz constant without loss of generality since it can be absorbed into the definition of the similarity metric. Individual fairness essentially caps the rate at which the predictions can change with respect to the input features. Individual fairness does not concern itself with the specific decisions $\hat{y}$, but rather the consistency with which they are made across individuals $|\mathrm{d}\hat{y}/\mathrm{d}\boldsymbol{x}|$. Individual fairness is a property of a mapping from input $\boldsymbol{x}$ to output $\hat{y}$ (or $y$), not a measure of how one mapping differs from another $\hat{y} - y$. In this sense, model accuracy and *individual fairness* are orthogonal. Individual fairness cares only how similar people are, not how they are ranked, the latter is determined by maximising utility. Rather than relying on the existence of a potentially untrustworthy ground truth data set, individual fairness requires the definition of a similarity metric to determine what is fair.

In summary, according to individual fairness the model mapping must be continuous and the smaller the model gradient (based on the similarity metric, $\mathrm{d}f/\mathrm{d}\boldsymbol{x}$), the more similarly individuals are treated. If the gradient is zero, we have equality - all individuals are mapped to the same distribution over outcomes and we have satisfied the individual fairness constraint. Of course such a model wouldn't be much use, as it would not take into account the features $\boldsymbol{x}$ of the individuals in its predictions. Essentially, we have reduced the problem of training a fair model $f(\boldsymbol{x}, y)$ to one of constrained optimisation:

$$\min_{\theta} \left\{ \mathbb{E}_{(\boldsymbol{x},y) \in (X, \boldsymbol{y})} \mathbb{E}_{\hat{y} = f(\boldsymbol{x}, y; \theta)} \left[ \mathcal{L}(X, \boldsymbol{y}, \hat{y}) \right] \right\},$$

$$\text{such that} \quad d_{\mathcal{P}(\mathcal{Y})}(f(\boldsymbol{x}_i, y), f(\boldsymbol{x}_j, y)) \leq d_{\mathcal{X}}(\boldsymbol{x}_i, \boldsymbol{x}_j) \quad \text{and} \quad f(\boldsymbol{x}_i, y) \in \mathcal{P}(\mathcal{Y}) \quad \forall \, \boldsymbol{x}_i, \boldsymbol{x}_j \in X.$$

It might be argued that individual fairness has not provided an answer to the problem, it has instead created a new one that's equally hard. However, in this new formulation, the importance of consistency in decision making is explicit. This separation of the tasks of maximising utility and ensuring fairness means they can be completed independently by two separate stakeholders. So a trusted party (regulator, auditor, risk manager) could determine the similarity function. The definition could be open, transparent and subject to debate and improvement over time as laws and culture evolve. The similarity metric essentially provides a mechanism through which to express a world view. In spirit, this characterization of (individual) fairness aligns with the arguments of Friedler et al. (2016); Binns (2019).

Conceptually, we can think of individual fairness as group fairness (more specifically statistical parity since it is interested in outcomes only), but in the limit as the subgroup size tends to one and $Z \to X$. Here, the outcomes are allowed to vary with the features, but within some limit which can be dependent on the feature. In theory, if $X$ includes sensitive features, we could set this limit to zero for those features by specification in the similarity metric, though in practice zeros might complicate things if we are for example trying to learn representations Lahoti et al. (2019). Thus, individual fairness can be viewed as an extension of group fairness rather than being in contention with it; faced with observations to the contrary, one would be wise to investigate further.

While randomness in judicial decisions might be contentious, in other contexts it might be desirable, or even necessary, to create a competent model. Indeed the most capable conversational AIs today produce randomised responses to prompts OpenAI (2023). In 2017, Google translate was consistently translating genderless languages (Finnish, Estonian, Hungarian, Persian, and Turkish) using gender-stereotyped pronouns in all cases (for example he is a doctor and she is a nurse) Caliskan et al. (2017). Randomised predictions would sometimes return "she is a doctor". Arguably, this would be fairer, since female doctors are erased by this technology. In 2019 almost 49% the doctors in the 37 countries in the Organization for Economic Cooperation and Development (OECD) were reportedly female OECD (2021). The problem that we identify here is not exclusive to gender. There are often multiple reasonable translations of the same sentence, even without gender associations. By returning only the most probable answer, the technology hides uncertainty and reduces diversity in its output. In 2018 Google introduced some fixes Kuczmarski (2018), but the solution (returning multiple translations) is not generalised beyond gender. Though perhaps not the best solution here, randomisation in predictions is one way to improve diversity in output and thus representation, which is where choosing the Bayes optimal solution fails spectacularly.

# 4 INDIVIDUAL COST AS AN EXTENSION OF EXPECTED RISK

In this section we describe the individual fairness metric used to evidence the trade-off between group and individual fairness Speicher et al. (2018), which we term *individual cost* to avoid confusion. Individual cost is calculated in two steps. First we must map model predictions $\hat{y}_i$ to the individual benefit $b_i$. Second, we must choose an an inequality index $I(\boldsymbol{b})$ to calculate how unequally benefits $\boldsymbol{b} = (b_1, b_2, ..., b_n)$ are distributed among the population consisting of $n$ individuals.

## 4.1 MAPPING PREDICTIONS TO BENEFITS

To map predictions to benefits in this case, authors suggest using one plus the error, that is, $b_i = \hat{y}_i - y_i + 1$. For a binary classifier, the error is either zero, or plus or minus one. Adding one to the error means the benefits are always zero or positive, a requirement for measuring inequality. In addition, at least one individual must benefit, so that the total benefit is positive. Here we assume that $y = 1$ is the advantageous outcome, in which case we can think of the benefit from the perspective of the individual as luck. A false positive, is the luckiest prediction ($b_i = 2$), a false negative the unluckiest ($b_i = 0$) and an accurate prediction is between the two ($b_i = 1$). More generally, authors suggest the benefit can be any function of $\hat{y}$ and target $y$, ignoring $\boldsymbol{x}$ ensures anonymity. For a classifier, the benefit function can be expressed as a matrix. $b_{ij} = \text{benefit}(\hat{y} = i, y = j)$.

*Individual cost* appears to represent a simplified measure of *individual fairness*. Rather than comparing individuals by their features, it compares them by their outcome - "treating individuals deserving similar outcomes similarly" Speicher et al. (2018). Clearly this metric is not the same as individual fairness as described in the previous section. Individual fairness does not trust the target at all while individual cost relies on it entirely. Individual cost is a property that compares two mappings (the prediction and the target) while individual fairness is a property of a single mapping.

## 4.2 MEASURING INEQUALITY WITH GENERALISED ENTROPY INDICES

There are many indices that measure inequality differently, but they all share the following properties:

- **Symmetry:** The function is symmetric in benefits, so that changing their order does not change the value of the index.
- **Zero-normalisation:** They are minimised with a value of zero, only for uniformly distributed benefits;
- **Transfer principal:** Transferring benefits from richer to poorer, always decreases the value of the index, provided the rich and poor don't switch places in their ranking as a result of the transfer; and
- **Population invariance** The measure is independent of the size of the population. It depends only on the distribution.

In addition to satisfying the above properties, generalised entropy indices are also *scale invariant*, that is the value of the index does not change under a constant scaling of the benefits.

For benefits $\boldsymbol{b} = (b_1, b_2, ..., b_n)$ with mean benefit $\mu$, the generalised entropy index can be written as,

$$I_\alpha(\boldsymbol{b}) = \frac{1}{n} \sum_{i=1}^n f_\alpha \left( \frac{b_i}{\mu} \right) \qquad \text{where} \qquad f_\alpha(x) = \begin{cases} -\ln x & \text{if} \quad \alpha = 0 \\ x \ln x & \text{if} \quad \alpha = 1 \\ \dfrac{x^\alpha - 1}{\alpha(\alpha - 1)} & \text{otherwise.} \end{cases} \tag{1}$$

The generalisation parameter $\alpha$ controls the weight applied to benefits at different parts of the distribution. Since benefits can be zero for the proposed choice of benefit function, we must choose a value of $\alpha > 0$. For example, Speicher et al. (2018) choose $\alpha = 2$ for their experiments.

Generalised entropy indices are the complete single parameter ($\alpha$) family of inequality indices with the additional property of subgroup decomposability Shorrocks (1980). This means that for any partition of the population into subgroups, we can additively decompose the index into a between-group component and a within-group component. The between group component is the contribution from variations in the mean benefit, between subgroups. The within-group component is the contribution from the variation in individual benefits, *within* the subgroups. If we think of the predictive system as a means for distributing benefits and we are able to map model predictions to benefits, we can use inequality indices to measure how unfair the algorithm is in its distribution of those benefits. If we use generalised entropy indices, we can think of the inequality index as a measure of individual unfairness and the between group component as a measure of group unfairness, allowing us to identify when trade-offs happen. Empirical evidence of the trade-off in Speicher et al. (2018) (shared with permission in the Appendix A.1) is provided for two data sets, both of which are binary classification problems. Index decomposition details can be found in Appendix A.2.

## 4.3 CONNECTION WITH UTILITY

The connection between measures of inequality and risk has been known for some time. For example, Atkinson exploited it to derive his own measure of inequality Atkinson (1970) over half a century ago. Note that calculating individual cost is much like calculating a cost sensitive loss. Here, the associated cost matrix is not constant, but rather depends on the mean benefit, $c_{ij} = \text{cost}(\hat{y} = i, y = j) = b_{ij}/\mu$. As our model performance changes, so does the mean benefit $\mu$, the associated costs $c_{ij}$ and index value $I(\boldsymbol{b})$. The mean benefit $\mu$ is always positive and so does not affect the relative size or ordering of the costs in the matrix. Thus, although mean benefit will affect the index value, it will not change the order of preference of different benefit distributions $\boldsymbol{b}$. We can understand the influence of the mean benefit on the index value as follows,

$$I_\alpha(\boldsymbol{b}) = \mathbb{E}\left[ f_\alpha \left( \frac{\boldsymbol{b}}{\mu} \right) \right] = \frac{\mathbb{E}[f_\alpha(\boldsymbol{b})] - f_\alpha(\mu)}{\mu^\alpha}.$$

If we define $b_i = \mathbb{P}(\hat{y}_i = y_i)$ and choose $\alpha = 0$, the index behaves like the cross entropy loss. If we instead have a regression problem we can define $b_i = \hat{y}_i - y_i$ and choose $\alpha = 2$, then the index behaves like the mean squared error. So, we can see that in some cases, this generalised metric behaves like known measures of expected risk. Individual cost might then be viewed as an extension of expected risk (or luck), since not all the possible metrics that can be constructed via the recipe, would result in a valid or meaningful objective function. For example, one such requirement is that of Fisher consistency. That is, we want a model to be an unbiased estimator of the target,

$\mathbb{E}(\hat{y}) = \mathbb{E}(y)$. For this we need the minima of our objective function to coincide with the expected value of the target. For a binary classifier with $b_i = \mathbb{P}(\hat{y}_i = y_i)$, $\alpha = 0$ and $\alpha = 2$, are the only two values for which this is true Buja et al. (2005).

The parameter $\alpha$ determines how fast the contribution to the index grows as a function of the benefit. For $\alpha > 1$ the contribution to the index grows faster than the benefit (prioritising equality among the lucky), and slower for $\alpha < 1$ (prioritising equality among the unlucky). Thus arguably, we should choose a value of $0 < \alpha < 1$. Finally, for the values $\alpha = 0$ and $\alpha = 1$, the within-group component is a true weighted average of the index values for the subgroups, since the coefficients sum to one. For $\alpha \in (0, 1)$ the coefficients sum to less than unity, being the sum minimised for $\alpha = 1/2$. For $\alpha > 1$, the coefficients sum to more than unity.

Though the original authors specify no constraints regarding the benefit matrix, we argue that there should be some, much like for the cost matrix. In particular, if an accurate prediction is more beneficial than an incorrect prediction (the leading diagonal dominates), then the index is simply a cost sensitive expected risk which always decreases with increasing accuracy. We can assume then, without loss of generality, that $\hat{y} = 1$ is the advantaged outcome. In this case, the second row of the benefit matrix $b_{ij} = \text{benefit}(\hat{y} = i, y = j)$ must dominate the first, $b_{10} > b_{00}$ and $b_{11} > b_{01}$. Since the index is scale invariant we can fix the value $b_{11} = 1$ (akin to fixing the unit of account). Since we know that a false negative is the least lucky outcome, setting this value to zero establishes the baseline. Although increasing all the benefits in the matrix by a fixed amount, will no doubt decrease inequality (a rising tide lifts all boats), it will not change the ranking of different benefit distributions. Then we can write the benefit matrix most generally as, $b_{ij} = ((b_-, 0), (b_+, 1))$ where $b_+ > b_- > 0$.

## 5   INDIVIDUAL COST ANALYSIS

In this section we elucidate further the behaviour of individual cost. To simplify the problem, we assume that accurate predictions are equally lucky. Then we have $b_- = 1$ with $b_+ > 1$. $b_+$ tells us how much luckier a false positive is compared to an accurate prediction. The value of the inequality index depends on the mean benefit $\mu$, or equivalently the total benefit $B = n\mu$. As the model performance changes, so does the mean benefit and thus the associated costs. The mean benefit $\mu$ is always positive and so does not affect the relative size or ordering of the costs in the matrix, but can still impact the relative preference of different predictions. Crucially, specifying $b_+ > b_-$ means that making a more accurate prediction might not always reduce the value of the index.

### 5.1   THE INDEX AS A FUNCTION OF MODEL PERFORMANCE METRICS

Individual cost can be written as a function of two model performance related parameters; prediction accuracy $\lambda$ and the mean benefit $\mu$. Recall, the latter is just the expected prediction error plus one $\mathbb{E}(\hat{Y} - Y) + 1$.

**Theorem 1.** *For the benefit function $b_i = \hat{y}_i - y_i + 1$ we can rewrite the generalised entropy index as:*

$$I_\alpha(\mu, \lambda) = \begin{cases} \left(1 - \dfrac{\lambda}{\mu}\right) \ln b_+ - \ln \mu & \text{if} \quad \alpha = 1 \\ \dfrac{1}{\alpha(\alpha - 1)} \left[ \left(\dfrac{b_+}{\mu}\right)^{\alpha - 1} - \dfrac{(b_+^{\alpha-1} - 1)}{\mu^\alpha}\lambda - 1 \right] & \text{if} \quad \alpha > 0. \end{cases} \tag{2}$$

See Appendix B for the proof.

Equation (2) shows that for fixed $\mu$, $I_\alpha(\mu, \lambda)$ is a linearly decreasing function of the accuracy $\lambda$. For $b_+ = 2$, the mean benefit $\mu$, gives us an indication of the relative number of false positive to false negative errors made by the model; it tells us if the model $\hat{y}$ is over or underestimating the target $y$ on average. From another point of view, it quantifies the amount of skew in the distribution of errors. $\mu < 1$ indicates positive skew (and vice versa for $\mu > 1$). Figure 1 provides visual illustrations of benefit distributions with different mean benefits $\mu$. When the mean benefit is one (as in the central figure), the distribution has no skew; it is symmetric.

Both $\lambda$ and $\mu$ are constrained. For any reasonable binary classifier, model accuracy is bounded, $0.5 \leq \lambda \leq 1$. The total number of benefits $B$, is minimized when all errors are false negatives and maximised when all errors are false positives. For a model with accuracy $\lambda = n_c/n$, the total benefit $B$, must satisfy the following bounds,

$$n_c \leq B \leq n_c + b_+(n - n_c) = b_+ n + (1 - b_+)n_c.$$

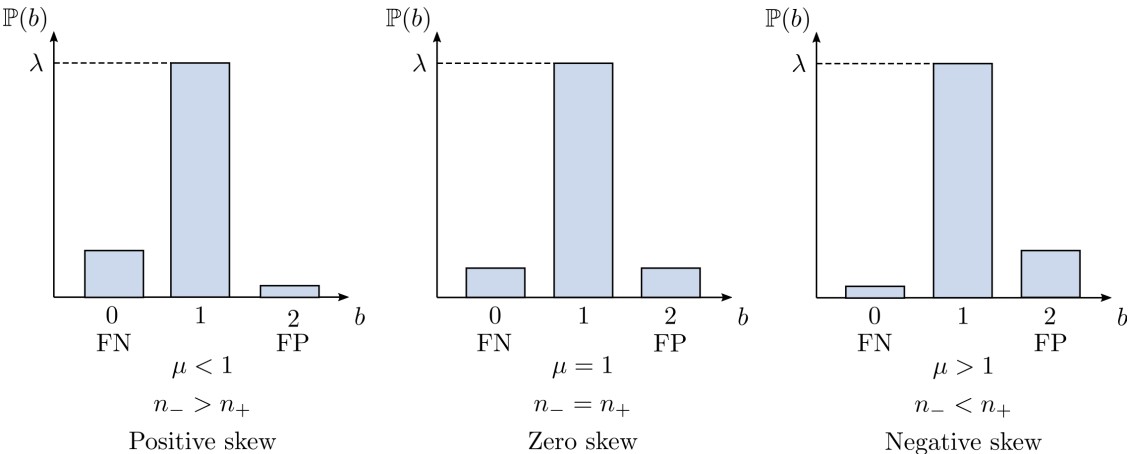

Figure 1: Characterization of benefit distributions with different mean benefits $\mu$.

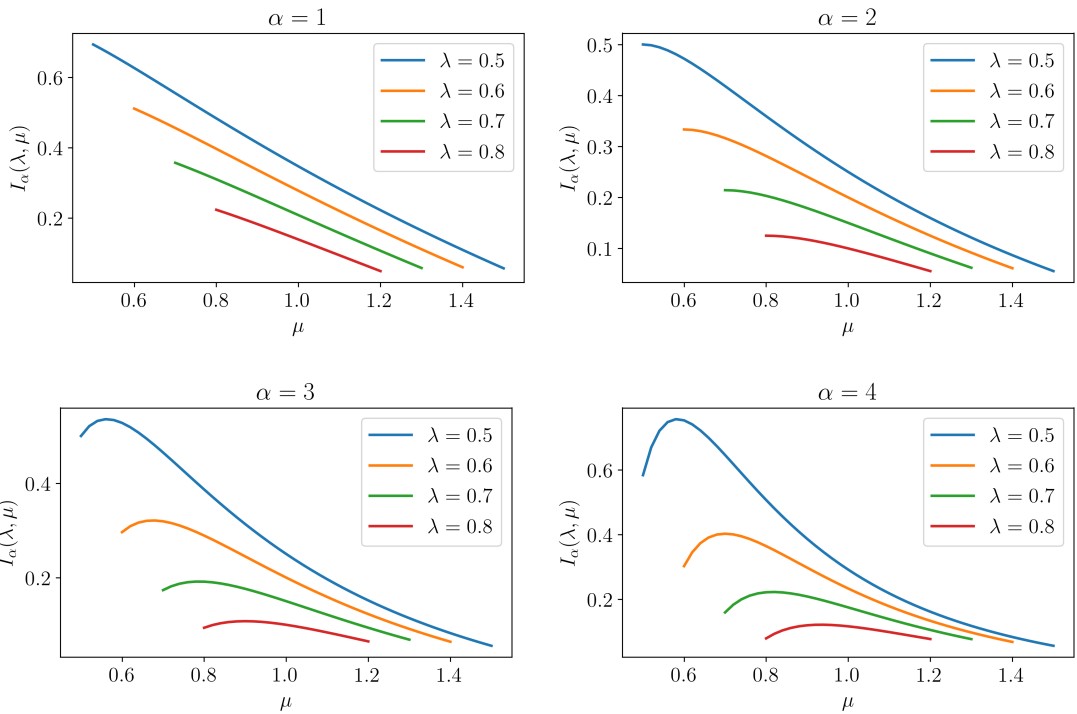

Figure 2: Generalised entropy index $I_\alpha \left( \mu, \lambda \right)$ as a function of $\mu$ for varying $\lambda$ and fixed $\alpha$ and $b_+ = 2$.

Since, $B = n\mu$, we must have $\lambda \leq \mu \leq b_+ + (1 - b_+)\lambda$. Thus, $I_\alpha(\mu, \lambda) : ([\lambda, b_+ + (1 - b_+)\lambda], [0.5, 1]) \mapsto \mathbb{R}_{\geq 0}$. As model accuracy $\lambda$ increases, the range of possible values the mean benefit $\mu$ can take, decreases. The domain is then a triangle. When $b_+ = 2$, the domain is an isosceles triangle. In Figure 2, we provide a side-view of the index surface for the case $b_+ = 2$.

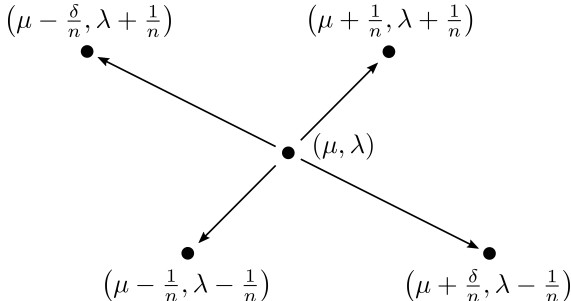

Figure 3: Visualisation of the local domain space for given $\mu$, $\lambda$ and $n$, assuming the point $(\mu, \lambda)$ is not on an edge.

## 5.2   WHEN DOES INCREASING THE ACCURACY REDUCE THE INDEX?

Let us denote the *cost of an error* as,

$$\Delta I_\alpha^\pm(\boldsymbol{b}) = I_\alpha(\boldsymbol{b}^\pm) - I_\alpha(\boldsymbol{b}).$$

Here $\boldsymbol{b}^\pm$ differs from $\boldsymbol{b}$ by one prediction only, containing one less correct prediction, and one more erroneous one. For $\boldsymbol{b}^+$, the additional error is a false positive. For $\boldsymbol{b}^-$, the additional error is a false negative. An additional false negative error, reduces the total benefit $B$, by one; both the accuracy $\lambda$ and the mean benefit $\mu$ are reduced by $1/n$. An additional false positive error, increases the total benefit $B$, by $\delta = (b_+ - b_-)$; the accuracy $\lambda$ is once again reduced by $1/n$, but this time the mean benefit $\mu$ increases by $\delta/n$. The discrete grid of adjacent models that we can reach through a small change in the model (given $\mu$, $\lambda$ and $n$), is shown in Figure 3. Hence,

$$\Delta I_\alpha^\pm(\mu, \lambda; n) = I_\alpha\left(\lambda - \frac{1}{n}, \mu \pm \frac{\delta}{n}\right) - I_\alpha(\mu, \lambda). \tag{3}$$

**Theorem 2.**

$$\left.\begin{array}{lll} \Delta I_\alpha^-(\mu, \lambda; n) < 0 & \Rightarrow & \mu < h^-(\alpha, b_+)\lambda \\ \Delta I_\alpha^+(\mu, \lambda; n) < 0 & \Rightarrow & \mu > h^+(\alpha, b_+)\lambda \end{array}\right\} \tag{4}$$

*where,*

$$h^\pm(\alpha, b_+) = \begin{cases} \dfrac{(b_+ - 1)\ln b_+}{b_+ - 1 \mp \ln b_+} & if \quad \alpha = 1 \\ \dfrac{\alpha(b_+ - 1)(b_+^{\alpha-1} - 1)}{[(\alpha - 1)(b_+ - 1) \mp 1]b_+^{\alpha-1} \pm 1} & if \quad \alpha > 0, \alpha \neq 1. \end{cases} \tag{5}$$

See Appendix B for the proof.

**Theorem 3.** *For the benefit function $b_i = \hat{y}_i - y_i + 1$, the only kind of error which is ever preferable to a correct prediction under this benefit function is a false positive. This happens only when the mean benefit exceeds $h^+(\alpha, b_+)\lambda$. We note that for a model whose accuracy is greater than $\hat{\lambda}(\alpha)$, it is not possible for the mean benefit to exceed the required level. That is,*

$$\Delta I_\alpha^-(\mu, \lambda; n) > 0 \quad \forall \mu, \lambda, n$$
$$\Delta I_\alpha^+(\mu, \lambda; n) < 0 \quad \Rightarrow \quad \mu > h^+(\alpha, b_+)\lambda,$$

*where, $h^+(\alpha, b_+)$ is defined in Equation (5).*

See Appendix B for the proof.

We term the part of the domain for which the index is reduced by decreasing the accuracy, the *deviation region*. The deviation region is described as, $\mu > h^+(\alpha, b_+)\lambda$. We mark the deviation region on the contour plot for $I_\alpha(\mu, \lambda)$ in Figure 4. These results show that individual cost expresses a preference for false positives, which makes sense given that they increase the total benefit. This is inline with the results from the original authors who find that for both tested datasets (Adult and COMPAS), rejecting more individuals increases the index. See the Appendix A.1 for details. For reference, in Table 1, we provide some values of $\hat{\lambda}(\alpha)$ and $h^+(\alpha, b_+)$. Now that we have identified the deviation region we return to the original question. For which set of error distributions does increasing the accuracy reduce

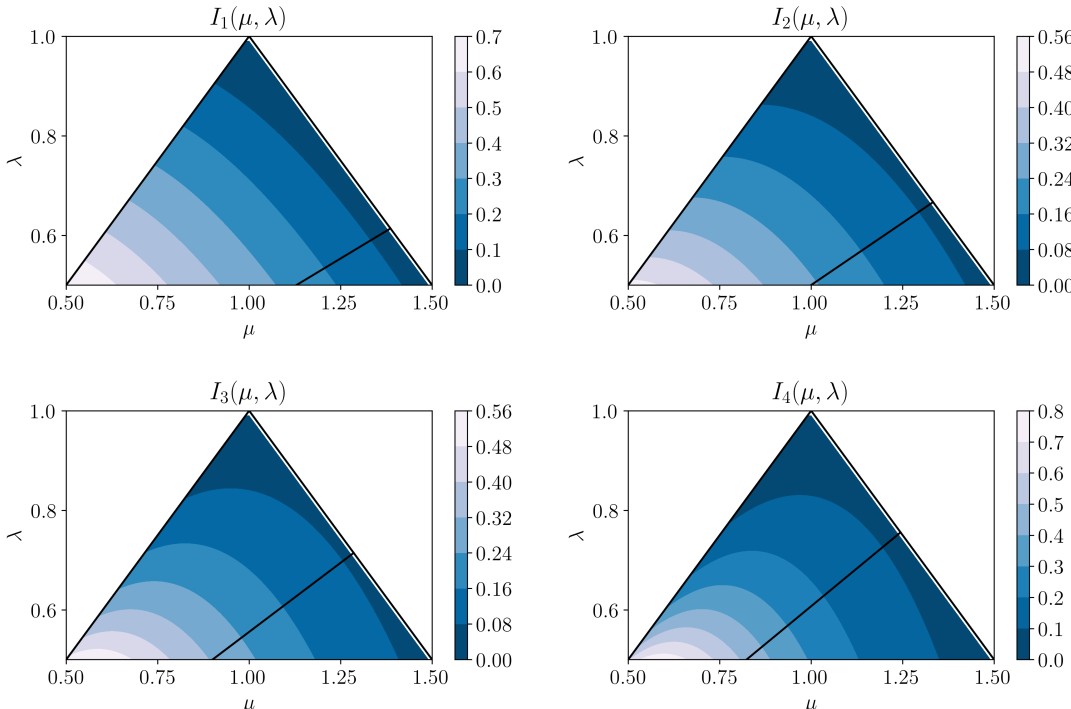

Figure 4: Contour plots showing $I_\alpha(\mu, \lambda)$ for different values of $\alpha$ and $b_+ = 2$.

Table 1: Reference thresholds that tell us when increasing the error rate must reduce the value of the index. This corresponds to the top of the triangular deviation regions shown in the contour plots in Figure 4.

| $\alpha$ | $\hat{\lambda}(\alpha)$[a] | $h^+(\alpha, b_+)$[b] |
|---|---|---|
| 1 | 61.4% | 2.26 |
| 2 | 66.7% | 2 |
| 3 | 71.4% | 1.8 |
| 4 | 75.6% | 1.65 |

a  We require $\lambda < \hat{\lambda}(\alpha)$ for the possibility that reducing the index value may not correspond to reducing the error rate. At $\lambda = \hat{\lambda}(\alpha)$, all the errors must be false positives to achieve the value of $\mu$ required for $\Delta I^+(\mu, \lambda; \alpha, n) < 0$.

b  We require $\mu > h^+(\alpha, b_+)\lambda$ for a false positive error to result in a reduction of the index value.

unfairness according to this metric? The analysis shows that this only happens when the ratio of false positives to correct predictions is sufficiently high (greater than half the number of accurate predictions for the specific parameter choices in the paper). Furthermore, such a ratio can only be achieved if the prediction accuracy is sufficiently low (66.7% for the specific parameter choices in Speicher et al. (2018)).

## 6  CONCLUSIONS

In this work we present arguments which resolve conflicting research on the nature of individual fairness Dwork et al. (2011). We clarify important defining features of individual fairness, framing it as an extension of group fairness, rather than acting in opposition to it. In particular, we emphasise the importance of the definition of individual fairness being orthogonal to utility. We review empirical evidence of the trade-off between group and individual fairness and derive a new representation for the associated individual fairness metric Speicher et al. (2018) (which we term individual cost). With this new representation we prove that individual cost is a function of model accuracy and express exactly when fairness and accuracy are inversely related. We conclude that empirical evidence does not support the existence of a trade-off between group and individual fairness but rather likely demonstrates the well known trade-off between fairness and utility.

## 7    Reproducibility Statement

For readers who wish to reproduce any part of this paper, all relevant resources are open source and can be found on GitHub. All proofs can be found in the Appendix.

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

## A  Individual Cost as an Extension of Expected Risk

### A.1  Empirical Evidence of the Trade-off Between Group and Individual Fairness

Figure 5 from Speicher et al. (2018) shows how the between-group component of the index $I_\beta^G(\boldsymbol{b}; \alpha)$ (solid line) and index value $I_\alpha(\boldsymbol{b})$ (dotted line) change with the acceptance threshold, for a variety of models. We see that the optimal thresholds for each metric (assumed to be measures of group and individual fairness respectively) do not coincide, supporting the theory of a trade-off between group and individual fairness.

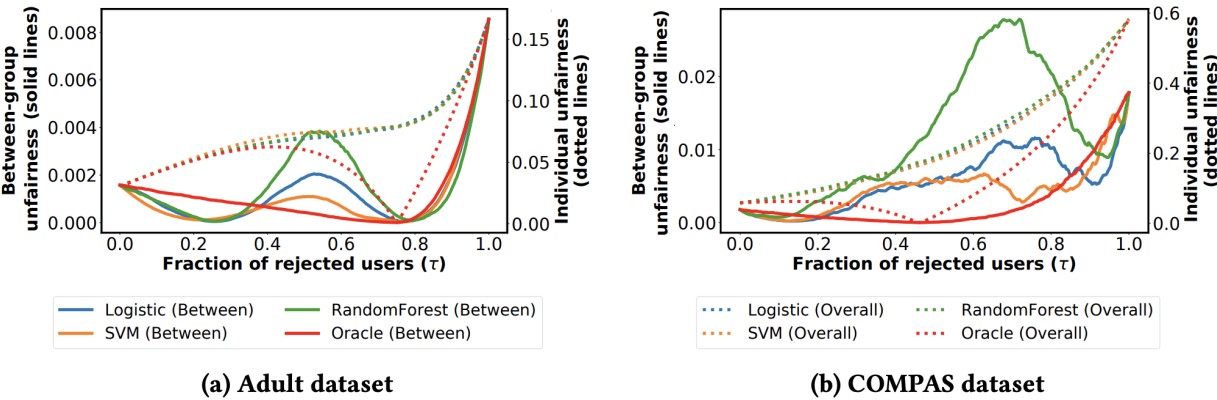

(a) Adult dataset             (b) COMPAS dataset

Figure 5: Between-group unfairness (solid lines) and overall unfairness (dotted lines) as a function of the decision ranking threshold ($\tau$) for various classifiers from Speicher et al. (2018).

### A.2  Generalised Entropy Index Decomposition

For any partition $G$ of the population into subgroups, the generalised entropy index $I$, is additively decomposable, into a within-group component $I_\omega^G$, and between-group component $I_\beta^G$,

$$I(\boldsymbol{b}; \alpha) = \frac{1}{n}\sum_{i=1}^{n} f_\alpha\left(\frac{b_i}{\mu}\right) = I_\omega^G(\boldsymbol{b}; \alpha) + I_\beta^G(\boldsymbol{b}; \alpha).$$

The within-group component is the weighted sum of the index measure for each subgroup

$$I_\omega^G(\boldsymbol{b}; \alpha) = \sum_{g=1}^{|G|} \frac{n_g}{n}\left(\frac{\mu_g}{\mu}\right)^\alpha I(\boldsymbol{b}_g; \alpha) \qquad \forall \alpha. \tag{6}$$

The between-group component is computed as the value of the index in the case where, each individual is assigned the mean benefit of their subgroup

$$I_\beta^G(\boldsymbol{b}; \alpha) = \sum_{g=1}^{|G|} \frac{n_g}{n} f_\alpha\left(\frac{\mu_g}{\mu}\right). \tag{7}$$

## B  Individual Cost Analysis

**Theorem 1.** *For the benefit function $b_i = \hat{y}_i - y_i + 1$ we can rewrite the generalised entropy index as:*

$$I_\alpha(\mu, \lambda) = \begin{cases} \left(1 - \dfrac{\lambda}{\mu}\right)\ln b_+ - \ln\mu & \text{if} \quad \alpha = 1 \\[2ex] \dfrac{1}{\alpha(\alpha-1)}\left[\left(\dfrac{b_+}{\mu}\right)^{\alpha-1} - \dfrac{(b_+^{\alpha-1} - 1)}{\mu^\alpha}\lambda - 1\right] & \text{if} \quad \alpha > 0. \end{cases} \tag{2}$$

*Proof.* Let's suppose the model makes $n_c$ correct predictions (in which case $b = 1$); $n_+$ false positive predictions (in which case $b = b_+$); and the remaining $n - n_c - n_+$ predictions are false negative (in which case $b = 0$). We can write the value of the index as,

$$I_\alpha(\boldsymbol{b}) = \frac{1}{n}\left[(n - n_c - n_+)f_\alpha(0) + n_c f_\alpha\left(\frac{1}{\mu}\right) + n_+ f_\alpha\left(\frac{b_+}{\mu}\right)\right].$$

From equation (1) we know,

$$f_\alpha(0) = \begin{cases} 0 & \text{for} \quad \alpha = 1 \\ \dfrac{-1}{\alpha(\alpha - 1)} & \text{for} \quad \alpha > 0, \end{cases}$$

$$f_\alpha\left(\frac{1}{\mu}\right) = \begin{cases} -\dfrac{\ln\mu}{\mu} & \text{for} \quad \alpha = 1 \\ \dfrac{1}{\alpha(\alpha - 1)}\left(\dfrac{1}{\mu^\alpha} - 1\right) & \text{for} \quad \alpha > 0, \end{cases}$$

$$f_\alpha\left(\frac{b_+}{\mu}\right) = \begin{cases} \dfrac{b_+(\ln b_+ - \ln\mu)}{\mu} & \text{for} \quad \alpha = 1 \\ \dfrac{1}{\alpha(\alpha - 1)}\left(\dfrac{b_+^\alpha}{\mu^\alpha} - 1\right) & \text{for} \quad \alpha > 0. \end{cases}$$

$$\Rightarrow \quad I_\alpha(\boldsymbol{b}) = \begin{cases} -\dfrac{(n_c + b_+ n_+)\ln\mu}{n}\dfrac{\ln\mu}{\mu} + \dfrac{b_+ n_+ \ln b_+}{n\mu} & \text{for} \quad \alpha = 1 \\ \dfrac{1}{\alpha(\alpha - 1)}\left(\dfrac{n_c + b_+^\alpha n_+}{n\mu^\alpha} - 1\right) & \text{for} \quad \alpha > 0. \end{cases}$$

Let us denote the accuracy of our model with $\lambda$. We have,

$$\lambda = \frac{n_c}{n} \quad \text{and} \quad \mu = \frac{n_c + b_+ n_+}{n} \quad \Rightarrow \quad \frac{b_+ n_+}{n} = \mu - \lambda.$$

Substituting completes the proof. $\qquad\square$

**Theorem 2.**
$$\begin{rcases} \Delta I_\alpha^-(\mu, \lambda; n) < 0 \quad \Rightarrow \quad \mu < h^-(\alpha, b_+)\lambda \\ \Delta I_\alpha^+(\mu, \lambda; n) < 0 \quad \Rightarrow \quad \mu > h^+(\alpha, b_+)\lambda \end{rcases} \tag{4}$$

*where,*
$$h^\pm(\alpha, b_+) = \begin{cases} \dfrac{(b_+ - 1)\ln b_+}{b_+ - 1 \mp \ln b_+} & \text{if} \quad \alpha = 1 \\ \dfrac{\alpha(b_+ - 1)(b_+^{\alpha-1} - 1)}{[(\alpha - 1)(b_+ - 1) \mp 1]b_+^{\alpha-1} \pm 1} & \text{if} \quad \alpha > 0, \alpha \neq 1. \end{cases} \tag{5}$$

*Proof.* Equation (2) provides an expression for $I_\alpha(\mu, \lambda)$. Substituting for $\lambda$ and $\mu$ in the case $\alpha = 1$ gives,

$$I_\alpha\left(\mu \pm \frac{\delta}{n}, \lambda - \frac{1}{n}\right) = \left[1 - \left(\frac{\lambda}{\mu} - \frac{1}{n\mu}\right)\left(1 \pm \frac{\delta}{n\mu}\right)^{-1}\right]\ln b_+ - \ln\mu - \ln\left(1 \pm \frac{\delta}{n\mu}\right).$$

For $\alpha > 0$, we get,

$$I_\alpha\left(\mu \pm \frac{\delta}{n}, \lambda - \frac{1}{n}\right) = \frac{1}{\alpha(\alpha - 1)}\left[\left(\frac{b_+}{\mu}\right)^{\alpha-1}\left(1 \pm \frac{\delta}{n\mu}\right)^{1-\alpha} - \frac{(b_+^{\alpha-1} - 1)}{\mu^{\alpha-1}}\left(\frac{\lambda}{\mu} - \frac{1}{n\mu}\right)\left(1 \pm \frac{\delta}{n\mu}\right)^{-\alpha} - 1\right].$$

We showed earlier that we must have, $\lambda \leq \mu \leq b_+ + (1 - b_+)\lambda$, in addition, any reasonable model should satisfy $0.5 \leq \lambda \leq 1$. We deduce that we must have $0.5 \leq \mu \leq 1.5$ and so $\mu = O(1)$. Then for large $n$, we can be sure that $n\mu$ is large and its reciprocal $\epsilon = 1/(n\mu)$ is small. For large $n$, we can write the cost of an error as

$$\Delta I_\alpha^\pm(\mu, \lambda; n) = \xi_\alpha(\mu, \lambda)\epsilon + O(\epsilon^2)$$

where,

$$\xi_\alpha(\mu, \lambda) = \begin{cases} \left(1 \pm \dfrac{\delta\lambda}{\mu}\right)\ln b_+ \mp \delta & \text{if} \quad \alpha = 1 \\ \dfrac{1}{\alpha(\alpha-1)\mu^{\alpha-1}}\left[\left[\left(1 \pm (1-\alpha)\delta\right)b_+^{\alpha-1} - 1\right]\mu \pm \alpha\delta(b_+^{\alpha-1} - 1)\lambda\right] & \text{if} \quad \alpha > 0. \end{cases}$$

$\square$

**Theorem 3.** *For the benefit function $b_i = \hat{y}_i - y_i + 1$, the only kind of error which is ever preferable to a correct prediction under this benefit function is a false positive. This happens only when the mean benefit exceeds $h^+(\alpha, b_+)\lambda$. We note that for a model whose accuracy is greater than $\hat{\lambda}(\alpha)$, it is not possible for the mean benefit to exceed the required level. That is,*

$$\Delta I_\alpha^-(\mu, \lambda; n) > 0 \quad \forall \mu, \lambda, n$$
$$\Delta I_\alpha^+(\mu, \lambda; n) < 0 \quad \Rightarrow \quad \mu > h^+(\alpha, b_+)\lambda,$$

*where, $h^+(\alpha, b_+)$ is defined in Equation (5).*

*Proof.* **False negatives**

$$h^-(\alpha, b_+) = \begin{cases} \dfrac{(b_+ - 1)\ln b_+}{b_+ - 1 + \ln b_+} & \text{if} \quad \alpha = 1 \\ 1 - \dfrac{(2 - b_+)b_+^{\alpha-1} + \alpha(b_+ - 1) - 1}{[(\alpha-1)(b_+ - 1) + 1]b_+^{\alpha-1} - 1} & \text{if} \quad \alpha > 0, \alpha \neq 1 \end{cases} \tag{8}$$

Equation 8 reveals that $h^-(\alpha, b_+)$ is a strictly increasing function of $\alpha$, for $\alpha > 0$ (since $\alpha b_+^{\alpha-1}$ dominates $\alpha$). In addition, we can see that $h^-(\alpha, b_+) \to 1^-$ as $\alpha \to \infty$. For a plot of $h^-(\alpha, b_+)$ see Figure 6 in the Appendix. Earlier

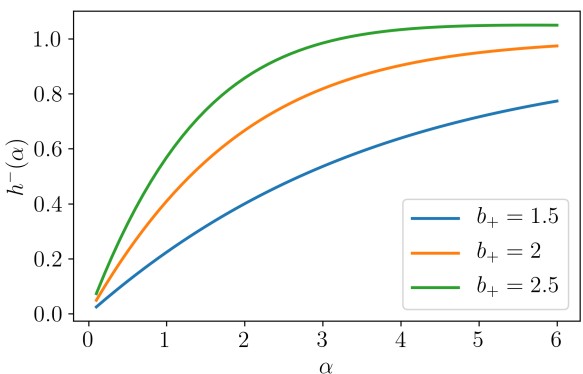

Figure 6: $h^-(\alpha, b_+) = 1 - \dfrac{(2 - b_+)b_+^{\alpha-1} + \alpha(b_+ - 1) - 1}{[(\alpha-1)(b_+ - 1) + 1]b_+^{\alpha-1} - 1}$.

we showed that we must have $\mu \geq \lambda$. Then from Equation (4), for $\Delta I_\alpha^-(\mu, \lambda; n) < 0$ we need $h^-(\alpha, b_+) > 1$. Since $h^-(\alpha, b_+) < 1$ for all $\alpha > 0$, we know that making an additional false negative error, never decreases the value of the index.

**False positives**

$$h^+(\alpha, b_+) = \begin{cases} \dfrac{(b_+ - 1)\ln b_+}{b_+ - 1 - \ln b_+} & \text{if} \quad \alpha = 1 \\ \dfrac{\alpha(b_+ - 1)(1 - b_+^{1-\alpha})}{\alpha(b_+ - 1) - b_+ + b_+1 - \alpha} & \text{if} \quad \alpha > 0, \alpha \neq 1 \end{cases} \tag{9}$$

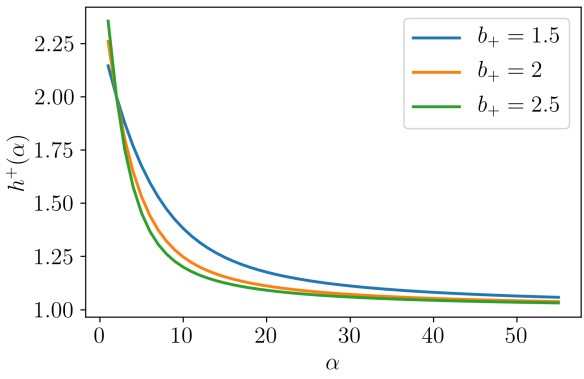

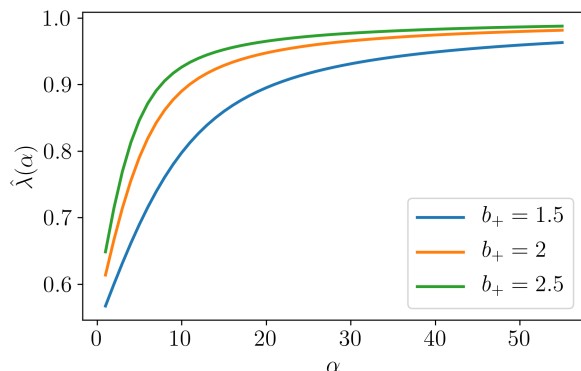

Figure 7: $h^+(\alpha, b_+) = \dfrac{\alpha(b_+ - 1)(1 - b_+^{1-\alpha})}{\alpha(b_+ - 1) - b_+ + b_+ 1 - \alpha}$

Figure 8: $\hat{\lambda}(\alpha) = \dfrac{b_+}{h^+(\alpha, b_+) + b_+ - 1}$.

Equation 9 reveals that $h^+(\alpha, b_+)$ is a decreasing function of $\alpha$, since $b_+ 1 - \alpha$ is a strictly decreasing function of $\alpha$. In addition, we can see that $h^+(\alpha, b_+) \to 1^+$ as $\alpha \to \infty$. Earlier we showed that we must have $\mu \le b_+ - (b_+ - 1)\lambda$. Then from Equation (4), for $\Delta I_\alpha^+(\mu, \lambda; n) < 0$ we need,

$$h^+(\alpha, b_+)\lambda < b_+ - (b_+ - 1)\lambda \quad \Leftrightarrow \quad \lambda < \hat{\lambda}(\alpha) = \frac{b_+}{h^+(\alpha, b_+) + b_+ - 1}.$$

From what we know about $h^+(\alpha, b_+)$, we can deduce that $\hat{\lambda}(\alpha)$ is an increasing function of $\alpha$, and $\hat{\lambda}(\alpha) \to 1^-$ as $\alpha \to \infty$. Since $\hat{\lambda}(\alpha) < 1$ for all $\alpha > 0$, we know there are indeed some circumstances, under which a false positive error, decreases the value of the index. For plots of $h^+(\alpha, b_+)$ and $\hat{\lambda}(\alpha)$, see Figures 7 and 8. □

