# OpenReview forum: "Individual Fairness as an Extension of Group Fairness"
_ICLR.cc/2024/Conference — Submitted to ICLR 2024_

### Official Review · Reviewer_RJ1C · 2023-10-30

**Soundness:** 3 good
**Presentation:** 1 poor
**Contribution:** 2 fair
**Rating:** 3
**Confidence:** 4

**Summary:**

This paper re-examines the definitions of individual fairness and individual cost. The authors claim that individual fairness can be considered as individual fairness by taking each individual example as a sole group. Then the authors proceed to formulate expressions for individual cost based on the cost utility. Through the visualizations the authors suggest the trade-off is between fairness and accuracy, rather than between individual fairness and group fairness.

---
**Update after rebuttal:**
Though the authors did not respond the reviewers separately, I read the authors' revisions and will retain my rating.

**Strengths:**

- This paper seems to be a comprehensive analysis for the relationship between individual fairness and individual cost. I appreciate it that the authors make very detailed interpretation of the notions of generalized entropy index and how they are derived in terms of the utility.
- I went through the derivation of the index functions and they seem to be correct.

**Weaknesses:**

- The paper is not well written. Notably, there is an overuse of possessive pronouns ("our features," "our model," "our xxx") throughout the text in the paragraph located at the end of page 3.
- Although the title “individual fairness as an extension of group fairness” is eye-bowling, this statement seems to be only related to Section - The rest of this paper is more than articulating the difference between individual fairness and individual cost.
- The definition of individual cost is deferred to the Appendix, making it hard to follow the main idea of the paper.

**Questions:**

- [Q1] How is the damped model output (the last equation at page 3) useful in Section 3.1? What is the error damping in your individual fairness problem?
- [Q2] How to interpret the conclusion at the end of page 4, i.e., individual fairness does not allow us to ignore uncertainty and choose the most probable response in all cases; it demands representation in model outputs. What is the representation referred to in this context?
- [Q3] In section 5, what is the implication of the variable $\alpha$ in the generalized entropy index?
- [Q4] Why is the index $I$ asymmetric for false positives and false negatives? Does that make sense?

---

### Official Review · Reviewer_1B79 · 2023-10-31

**Soundness:** 3 good
**Presentation:** 3 good
**Contribution:** 1 poor
**Rating:** 3
**Confidence:** 4

**Summary:**

This paper investigated the connections between group fairness and individual fairness. The main motivation is the stated inconsistency between the empirical evidence about the trade-offs between individual and group fairness, and the theoretical argument for the alignment between the two notions. This paper resolved this inconsistency in two ways. First, the authors showed that individual fairness can be framed as an extension of group fairness. Second, the authors looked into the early work providing empirical evidence, and clarified that the individual fairness notion studied in the early work captures fairness with respect to individual cost. Moreover, through detailed mathematical analysis, they argued that the empirical trade-off between group and individual fairness actually reflects the trade-off between fairness and utility.

**Strengths:**

Section 3 highlights useful properties of individual fairness. Section 5 provides detailed and clear mathematical examination of the generalized entropy index in terms of accuracy and mean benefit. These results reveal interesting patterns about the trade-off between accuracy and inequality.

**Weaknesses:**

The main weakness is that majority of this paper is about analyzing one early work, Speicher et al. (2018) Considering the large volume of fair ML literature, I am not convinced about the general usefulness of providing further explanation of results from one single paper. In addition, the current paper uses the conflicting empirical trade-off from Speicher et al. (2018) and the philosophical argument from Binns (2019) as the main motivation. I disagree with the claimed conflict. Binns (2019) adopted the individual fairness notion from Dwork et al. (2011), but Speicher et al. (2018) discussed in their Section 2.4 that their individual unfairness measure follows different principle from Dwork et al. (2011). Therefore, it is not surprising that the two papers concluded different results about individual-group fairness trade-offs.

About the analysis done in the current paper (Section 5), I see a disconnection between the derivation and the conclusion. The conclusion mentioned utility-fairness trade-off, but the utility definition is never clearly stated in the paper. It is therefore unclear what the utility-fairness trade-off is.

For the other set of contributions about connecting individual and group fairness, I find the observation that individual fairness can be interpreted as an extreme case of group fairness as group size approaches 1 to be straightforward.

**Questions:**

1.	How to define utility in the analysis of Section 5?

2.	The analysis in Section 5 uses the same model as the early paper. Do the techniques generalize to other formulation, e.g., different benefit definition, different inequality index?

3.	Are there other papers providing empirical evidence of the individual-group fairness trade-offs? If so, why did the authors study the results from Speicher et al. (2018) in particular?

---

### Official Review · Reviewer_iiHT · 2023-11-01

**Soundness:** 2 fair
**Presentation:** 2 fair
**Contribution:** 2 fair
**Rating:** 3
**Confidence:** 4

**Summary:**

The paper considers individual fairness (Dwork et al., 2011), and attempts to draw comparison between individual fairness and individual cost (Speicher et al., 2018), and that between individual fairness and group fairness. The paper presents how one can view individual fairness as an extension of group fairness, argues that individual fairness is orthogonal to utility in terms of definition.

**Strengths:**

The strength of the paper comes from the attempt to draw connection between several important considerations in algorithmic fairness research, including individual fairness, individual cost (as one type of individual fairness definition), group fairness, and utility.

**Weaknesses:**

The weakness of the paper comes from the unclear presentation of the problem formulation, and a lack of thorough literature review on the central topic. In particular, clarifications on the presented definition of individual fairness, which is different from the one formulated in Dwork et al. (2011), would be very helpful to understand the approach of analysis (detailed in Section __Questions__). Further discussions on other individual-level fairness notions, e.g., causal fairness notions, would help present a more comprehensive picture of the individual fairness studied in the literature.

**Questions:**

__Question 1__: the difference between the presented formulation of individual fairness, and the original definition presented in Dwork et al. (2011)

In the original individual fairness definition (Dwork et al., 2011), the prediction mapping is from input features to target variable ($M: V \rightarrow \Delta(A)$, where $V$ are features, and $A$ denotes outcome in their notation). In this work, the authors consider a generative modeling $\mathcal{X} \times \mathcal{Y} \rightarrow [0, 1]$, as in the definition of Lipschitz Continuity presented in Section 3. The definition claims to consider the mapping from "individuals to probability distribution over" outcome $Y$, then why $\hat{y} = f(x, y)$ take both $x$ and $y$ as input? If instead $f(x, y) \in \mathcal{P}$ (as specified in the paper), the output of $f(\cdot)$ is a distribution over outcome, instead of predicted value? The mathematical formulation is very confusing.

__Question 2__: the analyzing approach of individual fairness, where "error in features" are introduced

In Section 3.1, individual fairness as error damping, a Taylor expansion is presented. Following the above question, I am not sure how to parse this expansion. On the one hand, only the slight variation in features $x$ is included and only the partial derivative $\frac{\partial f}{\partial x}$ is included in the expansion, which seems to suggest that a discriminative (instead of generative) modeling is of interest. On the other hand, when both $x$ and $y$ are considered in the multivariate function $f(x, y)$, it is not reasonable to assume $\frac{\partial f}{\partial y}$ to always be 0 for any $(\tilde{x}, y)$ combination. I am having difficulty parsing the analyzing approach, especially considering the fact that randomness/error $\epsilon$ plays an important role in the presented discussion on individual fairness.

__Question 3__: what are the implications of the presented theorems?

Readers can benefit from more discussions on the implication of presented Theorems 1 -- 3. How to parse them and connect them to the discussion of individual fairness?

__Minor typo__:

- Page 3: Section 3.1 "for any error [in? or related to?] our features"

---

### Author Response · Authors · 2023-11-23
**Summary of changes**

Dear reviewers,

First we would like to express our gratitude for your valuable time and patience in reviewing our paper. We especially appreciated the very specific nature of the questions asked which helped us to identify that which was missing, superfluous or poorly placed; and hopefully substantially improve the clarity and quality of the work. In reviewing the revised paper you’ll find that we have addressed all the feedback and answered your questions. In particular we have,

- Updated the introduction to be more specific to the questions answered by the paper.
- Added some references to works that relate to causal inference.
- Added discussion around trade-offs and other related work.
- Amended our notation in the definition of individual fairness so that the model maps to a function over outcomes, as defined by Dwork et al.
- Removed irrelevant material on error damping (this was relevant to the impact for regression and should not have been included) which understandably caused confusion - apologies.
- Removed theorem 2 as was not relevant to the question (we answer in the paper) around the behaviour of individual cost; that is, when it behaves like expected risk and when it does not.
- Added section 4.3 explaining the relationship between _individual cost_ and _utility_ (or rather expected risk). It also places constraints around what are essentially free parameters in the original paper. We discuss the properties of the metric for specific choices of the generalisation parameter $\alpha$ and why the index is asymmetric. We also explain why, like the cost matrix in cost sensitive learning, the benefit function/matrix has two degrees of freedom. Individual cost can then be viewed as a measure of _expected luck_.
- We provide an analytical representation for individual cost which generalises over all but one of the degrees of freedom in the benefit matrix revealing the behaviour under the assumption that accurate predictions are equally _lucky_.
- Tidied up the language and explanations around _utility_, expected risk/cost, accuracy, etc.
- Added some context discussing representation and diversity in output through randomisation.
- Replaced possessive pronouns with a more appropriate convention.
- Fixed typos.

We hope you find the revised paper to be a more enjoyable and better justified read.

Yours faithfully.

---

### Meta-Review · Area_Chair_sx6p · 2023-12-06

**Metareview:**

This paper compares individual fairness and group fairness via individual cost, based on analytical and empirical approaches.

Strenghths: reviewers found the general approach (attempting to connect individual fairness and group fairness) interesting. The authors did a great job trying to respond to reviewers' questions and preparing a revision.

Weaknesses: reviewers were generally mildly excited about the results and would hope to see more comparisons with other notions of individual fairness especially given the large literature. In addition, multiple presentational issues were raised. Unfortunately, the issues remains after the rebuttal

**Justification For Why Not Higher Score:**

As suggested by the reviewer, this can be a great work if other notions of individual fairness can be considered. My personal thought is that this is an interesting paper but not significant enough yet (in its current form)

**Justification For Why Not Lower Score:**

N/A

---

### Decision · Program_Chairs · 2024-01-16

Reject